# Going with the floe: tracking CESM Large Ensemble sea ice in the Arctic provides context for ship-based observations

Alice K. DuVivier[1], Patricia DeRepentigny[2,3], Marika M. Holland[1], Melinda Webster[4], Jennifer E. Kay[2,5], Donald Perovich[6]

[1]National Center for Atmospheric Research, Boulder, CO, 80307, USA
[2]Department of Atmospheric and Oceanic Sciences, University of Colorado Boulder, CO, USA
[3]Institute of Arctic and Alpine Research, University of Colorado Boulder, CO, USA
[4]University of Alaska Fairbanks, Fairbanks, AK, USA
[5]Cooperative Institute for Research in Environmental Sciences, University of Colorado Boulder, CO, USA
[6]Thayer School of Engineering, Dartmouth College, Hanover, NH, USA

*Correspondence to*: Alice K. DuVivier (duvivier@ucar.edu)

**Abstract.** In recent decades, Arctic sea ice has shifted toward a younger, thinner, seasonal ice regime. Studying and understanding this "new" Arctic will be the focus of a year-long ship campaign beginning in autumn 2019. Lagrangian tracking of sea ice floes in the Community Earth System Model Large Ensemble (CESM-LE) during representative "Perennial" and "Seasonal" time periods allows for understanding of the conditions that a floe could experience throughout the calendar year. These model tracks put into context a single year of observations, provide guidance on how observations can optimally shape model development, and how climate models could be used in future campaign planning. The modelled floe tracks show a range of possible trajectories, though a Transpolar Drift trajectory is most likely. There is also a small but emerging possibility of high-risk tracks, including possible melt of the floe before the end of a calendar year. We find that a Lagrangian approach is essential in order to correctly compare the seasonal cycle of sea ice conditions between point-based observations and a model. Because of high variability in the melt season sea ice conditions, we recommend *in-situ* sampling over a large range of ice conditions for a more complete understanding of how ice type and surface conditions affect the observed processes. We find that sea ice predictability emerges rapidly during the autumn freeze-up and anticipate that process-based observations during this period may help elucidate the processes leading to this change in predictability.

## 1 Introduction

In recent decades, sea ice in the Arctic Ocean has undergone rapid change (Serreze & Stroeve, 2015; Stroeve & Notz, 2018). Passive microwave satellite observations since 1979 show that Arctic sea ice extent has decreased in all months, and the twelve lowest September sea ice extents were recorded in the past twelve years (Richter-Menge et al., 2019). The reduced sea ice

cover has local effects on boundary layer clouds, temperature, and humidity, which can feedback on the sea ice evolution ( Kay & Gettelman, 2009; Boisvert & Stroeve, 2015; Morrison et al., 2018) and the large-scale atmospheric circulation (e.g., Alexander, 2004; Barnes & Screen, 2015; Deser et al., 2016).

Year-round *in-situ* observations are critical for understanding the coupled air-sea-sea ice processes over the remote Arctic
Ocean, but they pose enormous challenges. The Surface Heat Budget of the Arctic (SHEBA) project obtained year-round, process-based observations over sea ice when the Canadian Coast Guard icebreaker Des Groseilliers was frozen into the Beaufort Sea and drifted freely with the pack from October 1997 to October 1998 (Uttal et al., 2002). SHEBA observations have been immensely helpful for process-based understanding of the coupled system, and have been widely used to improve modelling of processes in the polar regions (e.g. Intrieri et al., 2002; Bromwich et al., 2009; Klein et al., 2009). Since the late
1990s when SHEBA occurred, there has been year-round sea ice loss (Stroeve & Notz, 2018), there is more first-year sea ice compared to multiyear ice (Maslanik et al., 2011; Nghiem et al., 2007), the pack has thinned substantially (Kwok, 2018; Kwok et al., 2009), and the melt season length has increased (Stammerjohn et al., 2012). Whether or not year-round coupled processes are similar in this new regime of young, thin, seasonal sea ice is an open question.

The international Multidisciplinary drifting Observatory for the Study of Arctic Climate (MOSAiC) experiment has been
designed to answer the question: what are the causes and consequences of an evolving and diminished Arctic sea ice cover? MOSAiC aims to assess coupled air-sea-sea ice processes as well as to investigate the impact on ecosystems and biogeochemistry of the changing system in order to answer MOSAiC's driving question and improve our understanding and modelling of polar processes in a changing climate (Dethloff et al., 2016). In autumn 2019, the icebreaker RV Polarstern will be frozen into the Siberian Arctic with the aim of traversing the Transpolar Drift current over the following year. Extensive
analysis using historical satellite data with the IceTrack Lagrangian approach (Krumpen et al., 2019) has been performed in order to identify MOSAiC's starting location, assist with logistical planning, and coordinate research efforts (Krumpen, 2019). Throughout the duration of the experiment dynamical sea ice forecasts, initialized with and assimilating the most up to date ice and weather data, will be performed for the particular MOSAiC track and are available online (https://sidfex.polarprediction.net/; coordinated by Helge Goessling). Analyses of an observationally-initialized ensemble
forecast can provide skilful forecasts for MOSAiC conditions and information about how long these forecasts are skilful before the system diverges from the initial state. These types of in-depth observational and observationally-initialized forecast analyses are necessary and important for a successful campaign (IASC, 2016).

In recent years there has been increasing awareness of the impact of internal climate variability on the possible range of sea ice conditions and the resulting representativeness of a single year of observations ( Swart et al., 2015; Jahn et al., 2016). In
this study, we use data from the Community Earth System Model (CESM) Large Ensemble (CESM-LE) project (Kay et al., 2015). The CESM-LE is an initial condition ensemble, meaning that each ensemble member represents one possible response of the climate system to the external forcing given inherent internal climate variability. The ensemble mean of the individual

model experiments represents the response to the changing external forcing, whereas the difference of each ensemble member from to the ensemble mean provides a measure of internal variability. Using the Lagrangian Ice Tracking System (LITS; DeRepentigny et al., 2016), we derive the tracks of virtual sea ice floes for each ensemble member and the evolution of floe conditions over a calendar year. This analysis is not equivalent to examining observational tracks over time. Observational tracks are affected by both internal variability and forced change. Instead, the CESM-LE is an ideal tool for disentangling the effects of internal climate variability from forced change on the conditions a field campaign might encounter.

The purpose of this study is not to provide a forecast for the particular sea ice conditions during MOSAiC. Instead, we use the likely starting condition determined by the MOSAiC planners to address three fundamental goals: (1) offer insight on the representativeness of MOSAiC observations given the range of internal climate variability; (2) provide guidance about what types of observations can best assist with model improvement and appropriate ways these observations can be used to improve climate models; (3) show how free running climate model simulations might assist with future campaign planning. In section 2, we describe LITS as well as the CESM-LE and observational data used in this study. Subsequently, section 3 describes the resulting Lagrangian tracks, sea ice conditions along the tracks, and initial-value predictability. We conclude and discuss resulting recommendations for both sea ice observations and models in section 4.

**2 Data and Methods**

The CESM-LE is a publicly available initial condition ensemble and is designed to assess the role of internal variability in the presence of forced change within the climate system. Each of the 30 CESM-LE members uses an identical code base and external historical and future climate forcing. The ensemble members are unique due to round-off level temperature differences ($10^{-14}$K) in the initial atmospheric conditions in 1920. Over time these round-off level differences lead to chaotic evolution in the climate system akin to the initial condition impact on weather forecasts (Lorenz, 1963). Therefore, the spread in individual CESM-LE members is generated solely by internal climate variability. Comparisons between ensemble members allow us to better understand and contextualize the range of possible floe tracks, sea ice conditions, variability, and predictability.

Climate models evolve freely, so they cannot and should not exactly represent the observed historical Arctic conditions. Therefore it is not possible to validate a model by comparing a single climate simulation to the observations, nor is it appropriate to directly compare the observations to an ensemble mean as by design the ensemble mean has damped internal climate variability (Kay et al., 2015). The CESM-LE well captures the Arctic sea ice historical state and trends, and the mean state compares best to the observations relative to other Coupled Model Intercomparison Project Phase 5 (CMIP5) experiments (Barnhart et al., 2015; Jahn et al., 2016). Few other CMIP5 models contribute multiple ensemble members and none had enough members to adequately quantify the internal variability in Arctic sea ice. In this way, the CESM-LE is unique because it does sufficiently represent the spread in conditions associated with internal climate variability (Jahn et al., 2016). The CESM-LE also reasonably represents the pattern and magnitude of Arctic sea ice thickness (Labe et al., 2018). The CESM-LE sea ice

motion patterns are similar to observations, though the Beaufort Gyre circulation is stronger than observed (DeRepentigny et al., 2016). Thus, due to its well represented Arctic sea ice mean state and variability and the availability of many ensemble members, the CESM-LE is an ideal global climate model for the following analyses.

For this study, 30 CESM-LE members provide daily sea ice concentration and velocity (u and v) fields. We also use satellite-derived sea ice velocity (Tschudi et al., 2016) and concentration (Meier et al., 2017) from 1988-2016, for a total of 28 year-long observationally-derived drift tracks. We use LITS (DeRepentigny et al., 2016) to track virtual sea ice floes starting from the point 85°N, 125°E on October 15 (as provided by Thomas Krumpen as a likely starting point for the MOSAiC campaign) for: (1) satellite-derived historical conditions, (2) Perennial Arctic model conditions, and (3) Seasonal Arctic model conditions. We obtained simulated along-track floe characteristics (e.g. sea ice thickness, snow thickness, turbulent heat fluxes) for each of the unique tracks by using a weighted average of all model grid cells within 50 km of the latitude and longitude provided by LITS on that day. CICE4, the sea ice model used in the CESM-LE, uses an ice thickness distribution to represent subgrid-scale heterogeneity which allows us to consider the predictability of concentration by thickness category. In the CESM-LE, we use five thickness categories that correspond to the following thickness ranges: 0-0.59 m, 0.6-1.39 m, 1.4-2.39 m, 2.4-3.59 m, 3.6+ m (Hunke & Lipscomb, 2008).

For this paper, the term "Seasonal" ("Perennial") corresponds to drifts from October 15, 2021 (1980) to October 15, 2022 (1981). The fundamental distinction between these regimes is the transition from old, thick, perennial sea ice to young, thin, more seasonal sea ice (Perovich, 2011). The purpose of including Perennial conditions, which no longer exist in the present-day Arctic, is to contrast with the Seasonal conditions in terms of how the mean state and variability, or spread in conditions, has changed over time. The specific years of model data were chosen based on model restart file availability needed to obtain daily values of all sea ice variables necessary for analysis as well as a clear representation and contrast of Perennial and Seasonal conditions. To evaluate the sensitivity of our results to the start location, we also ran LITS for a range of starting locations in the Siberian Arctic within +/- 5° latitude or longitude of the given starting point for MOSAiC (85°N, 125°E). While there were small changes in the floe track locations, the results were not significantly different from those presented here and are therefore not shown.

## 3 Results

### 3.1 Floe Tracks

To understand whether the shift from Perennial to Seasonal sea ice leads to changes in floe paths, we use LITS tracks to obtain statistical information relevant to planning an expedition's year-long drift. For the satellite-derived drifts, more recent years tend to have longer drift distances (Table 1), indicating that thinner ice may lead to longer travel. While both the Seasonal and Perennial CESM-LE mean track distances are longer than the satellite-derived tracks (indicating that model ice speeds are

faster than observations), Seasonal tracks tend to travel further than Perennial tracks (Table 1). Therefore, it is likely that an experiment like MOSAiC drifting in thin ice conditions will travel further than it would have when the sea ice was thicker due to observed faster drift speeds for thinner ice (Morison & Goldberg, 2012; Rampal et al., 2009; Tschudi et al., 2019). Additionally, we find that five Seasonal tracks (17%) melt before October 15 of the following year, with the earliest melt date on July 29 and the latest on September 22. Consequently, there is an emerging risk that the floe may melt out before the end of a calendar year.

Examining individual tracks (Fig.1, grey lines) provides perspective about paths the icebreaker might travel within a year, including high-risk paths. Figure 1 shows four different sectors in which the floe will end a year-long drift: the Russian Sector, the Canadian Sector, the Transpolar Drift sector, and the North Pole sector. Over the 1988-2015 period, the satellite-derived tracks most frequently end in the Transpolar Drift sector (46%; 13 tracks). In the first half of the observational record, prior to 2002, many tracks end near the North Pole sector (43%; 6 tracks) or enter the Russian sector (43%; 6 tracks). In later years, after 2002, the satellite-derived paths tend to shift toward the Canadian sector, though most ultimately end with a Transpolar Drift path (79%; 11 tracks). For the CESM-LE, the likely end points for tracks shift from the North Pole sector (63%; 19 tracks) in Perennial conditions to the Transpolar Drift sector (47%; 14 tracks) in Seasonal conditions. Therefore, with thinner sea ice, both the observations and model show an increased frequency of Transpolar Drift tracks. Of additional concern is the possibility that the track may enter a nation's exclusive economic zone (EEZ). In particular, if the experiment were to enter the Russian EEZ there may be an immediate cessation of all measurements, so understanding the likelihood of this occurrence in Seasonal sea ice conditions is important. Three satellite-derived tracks enter the Russian EEZ, while for the CESM-LE only one Seasonal and one Perennial track enter the Russian EEZ. This indicates that from this particular starting point a high-risk track into the Russian EEZ is unlikely even with the range determined by internal variability. The CESM-LE tracks tend to have higher probability of ending in the Canadian sector, which may be related to the position of the modelled centre of the Beaufort Gyre (DeRepentigny et al., 2016) that results in model tracks that tend to be shifted toward the Canadian sector.

We created maps showing the number of times each grid cell is visited by sea ice tracks over two-week periods throughout the year (Fig. 2). These maps are also used to identify "representative" tracks (Fig.1, coloured lines) by identifying locations with high track counts that also formed a continuous path – i.e. a path in which unphysical "jumps" in the track were not permitted. It is important to note this is not a forecast of the likely path MOSAiC will take, but instead is meant to represent a reasonable path given the individual tracks from ensemble members under the same climate forcing and how these may differ between Seasonal and Perennial conditions. While all representative tracks follow a Transpolar Drift trajectory, the representative Seasonal path is longer and shifted further towards the Canadian Arctic compared to the representative observed and simulated Perennial paths, which end further north (Fig. 1). The maps shown in Fig. 2 can also be used to inform the remote sensing community about the likelihood of when the field experiment might be observable by satellites. Tracks in close proximity to the North Pole are often not observable by most polar orbiting satellites due to orbit inclinations and instrument swath creating

gaps in coverage. These "pole holes" range in size (e.g., ~82.5°N for CloudSat; ~88°N for ICESat-2; ~89.3°N for AMSR-E) and are dependent on a satellite's orbit. Knowing when the floe is likely to be in these areas is valuable for planning and coordinating surface-based and airborne measurements to fill the high-latitude satellite "gap" (Fig. 2). Equally, knowing where and when the track is likely to emerge from satellite pole holes in spring after polar day has returned is valuable for planning

visible image acquisition, such as that from DigitalGlobe's WorldView satellites (https://www.satimagingcorp.com/satellite-sensors/), to support operational and scientific needs. All three representative tracks enter the satellite "gap" in December, but the difference in when the representative tracks exit the "gap" differs between September for observational and Perennial tracks and July for Seasonal tracks.

**3.2 Seasonal Floe Conditions and Variability**

The CESM-LE also provides guidance on the range of sea ice conditions and variability that may be encountered during a year-long expedition and whether these have changed over time. Individual ensemble members provide unique realizations of equally likely sea ice states, and the ensemble mean provides guidance on the most likely conditions.

The initial sea ice conditions are important both logistically and scientifically. Logistically, the initial ice state has implications on the icebreaker's ability to reach the desired destination and the fuel required. Scientifically, establishing an initial location

that has a mix of multiyear and first-year ice is important to ensure sampling of heterogeneous ice types. For Seasonal drifts, the initial ice coverage can have a range of possible concentration configurations, but for the starting location of 85°N, 125°E, the ensemble means shows that it is likely to be within the sea ice pack (Fig. 3a), though it may be near the sea ice edge (e.g. Ens. 2). The ensemble mean initial sea ice thickness is likely to be around 0.75 m (Fig. 3b), which is within the icebreaker's limitations. The ensemble mean ice age is around 1.75 years (Fig. 3c), indicating that there are likely to be multiyear floes at

this location. The variability in the initial sea ice conditions is unsurprisingly higher for Seasonal than Perennial conditions, which were likely to have extensive, thick (>2 m), multiyear (>4 year) ice at this location (Fig. 4). The increase in variability indicates that finding the ideal mix of ice conditions for MOSAiC is less certain because of the large variability in individual ensemble members sea ice states at the campaign initiation.

The along-track sea ice state throughout the year for each unique Seasonal and Perennial track provides information about the

seasonal evolution and variability in sea ice concentration and thickness. The sea ice concentration for all Seasonal and Perennial tracks is above 95% until about May 1, when the concentration begins to decrease (Fig. 5a, b). For Seasonal tracks, the initial ice thickness could range from 0.2 m to 2 m, though it is unlikely to exceed 3 m or fall below 1 m during the year-long drift (Fig. 5c). There is a notable increase in melt season (taken here as May 1 to September 15) ice concentration variability for Seasonal conditions compared to Perennial conditions. The average standard deviation for sea ice concentration

is more than double for Seasonal (5.5%) compared to Perennial (2.6%) tracks. In contrast, the standard deviation in sea ice thickness is higher for Perennial (0.54 m) compared to Seasonal (0.38 m) floes (Fig. 5d). The increased thickness variability

in the Perennial tracks is likely because the thick multiyear ice has a relatively weak negative ice thickness-ice growth rate feedback compared to thin ice cover. This feedback, in which thinner ice grows more rapidly because of increased heat conduction, damps ice thicknesss anomalies, and thus the ice thickness variability (Bitz & Roe, 2004; Goosse et al., 2018). This feedback is particularly strong in a thin ice regime.

In addition to placing observations into context, we also address the challenge of evaluating modelled sea ice using single point observations. Compared to the Lagrangian along-track sea ice concentration, the polar-cap (70-90°N) average concentration and variability are lower, and the change in the rate of decline occurs earlier (Fig. 5a, b). This difference is likely due to along-track positions remaining at higher latitudes (>80° N) where they receive less sunlight and typically melt later than sea ice would at lower latitudes (Bliss & Anderson, 2018). Because the timing of the seasonal cycles in pan-Arctic or polar-cap averages is not an accurate representation of the seasonal cycle for a given floe, future model studies comparing model output to MOSAiC observations should strive to use a Lagrangian comparison.

The variability as measured by the across ensemble-member standard deviation is also higher for Seasonal surface conditions that impact sea ice evolution – snow fraction, pond fraction, and albedo (Fig. 6). Unsurprisingly, in a warmer climate like that associated with Seasonal conditions, there tends to be higher pond fraction and lower snow cover fraction. Additionally, the standard deviation for snow depth is higher for Perennial (0.15 m) compared to Seasonal (0.38 m) tracks because most Seasonal tracks experience complete snow melt in July (Fig. 6d). In contrast, modelled Perennial tracks tend not to lose all their snow, which has been previously documented in CESM simulations (Light et al., 2015). The snow differences impact the ice concentration variability through positive albedo feedback (Goosse et al., 2018; Hall, 2004; Qu & Hall, 2007), and as a result of these surface conditions the Perennial tracks maintain a higher albedo throughout summer (Fig. 6a).

There are large differences in surface conditions during the melt season between the Seasonal and Perennial conditions, which results in the large differences in net shortwave radiation (Fig. 7a). The variability in the Seasonal energy and mass budget terms is much larger than the Perennial conditions (Fig. 7b, c, e, f). Seasonal tracks have more top melt initially compared to Perennial tracks but later differences in bottom melt become dominant (Fig. 7d). For Seasonal tracks the loss of snow increases surface shortwave absorption for both sea ice and the surrounding ocean, which leads to increases in top and bottom melt (Perovich et al., 2007).

These results of conditions and energy and mass budgets along the floe tracks are particularly useful for a two-way exchange of information: (1) they place context on a single year's worth of observations during a year-long field campaign, (2) remotely-sensed observations along the modelled drift tracks from the year before, during, and after the MOSAiC campaign can be collected to assess the magnitude of inter-annual variability of the modelled variables, and (3) the observations will help constrain the range in variability shown by the model for sea ice and surface conditions.

### 3.3 Floe Predictability

The CESM-LE also provides the opportunity to explore the initial-value predictability for the state of the expedition's sea ice floe. For campaign planning purposes, it is important when establishing an initial ice camp to be confident that the sea ice floes
will be sufficiently stable to prevent endangering personnel or equipment. By calculating the autocorrelation coefficient between the 30 unique initial floe conditions and the subsequent conditions each following month throughout the year, we are able to explore how long the initial sea ice state persists. This provides information on predictability of conditions, although notably there are other factors, such as the persistence of conditions that affect the sea ice state, that can give rise to predictability and are not accounted for in the autocorrelation analysis shown here.

For Seasonal tracks with a campaign start date of October 15 there is high and significant correlation for grid-cell mean sea ice thickness between the initial value and values well into the following year (Fig. 8a). While the autocorrelation is low for grid-cell mean sea ice concentration, the autocorrelation for concentration of ice in thicker categories is high, which reflects the high initial-value predictability of Arctic sea ice thickness (Blanchard-Wrigglesworth et al., 2011a, 2011b). The negative correlations for ice in category 2 (0.6-1.39 m) are due to the nature of the ice thickness distribution and indicates that ice that
is initially in this category is likely to move out of this category to thicker categories throughout the winter. The autocorrelation of sea ice variables for a Seasonal floe is typically lower compared to Perennial autocorrelations initialized on October 15 (Fig. 8b), suggesting lower predictability for Seasonal conditions. We tested other predictors (e.g. initial fraction of open water or sea surface temperature) for the sea ice floe's state at later dates during the campaign, but these were found to have low, statistically insignificant correlations, even from the initial week, and therefore were poor predictors (not shown).

The exact start date of the campaign is indefinable but may occur as early as October 1 or possibly as late as October 31. To understand how the predictability may change based on start date, we also evaluate the predictability characteristics for tracks initialized at the same location but on different dates. For Seasonal floes, an October 1 start date results in lower autocorrelations (Fig. 8c), while an October 30 start date has nearly identical autocorrelations (not shown) compared to an October 15 start date. In contrast, for a Perennial floe the autocorrelations are similar between October 1 and October 15 start
date (Fig. 8d). Because sea ice state predictors change rapidly during these two weeks, predictors used in Perennial conditions during October may not be appropriate in Seasonal conditions at the same calendar date depending at the stage of freeze-up. The evolution of differences in predictability characteristics that emerge during the autumn freeze-up are not yet well understood.

### 4 Discussion and Conclusions

The CESM-LE is a fully coupled global climate model with well represented Arctic sea ice mean state and variability. As an initial condition ensemble, the CESM-LE is a tool designed to explore the effects of internal climate variability and forced

change. Thus, by tracking modelled sea ice floes using the CESM-LE characteristic Perennial and Seasonal sea ice conditions, we have an ideal framework to meet our three goals: (1) put into context a single year's observations since MOSAiC will represent a single response of the climate system to forced change; (2) provide guidance about observations that can be used to improve climate models; (3) demonstrate how free running climate models might assist with future campaign planning.

Substantial work by MOSAiC planners has gone into both determining a starting location (85°N, 125°E) for the campaign and developing a forecast system for the campaign once it has initialized. This study is not intended to provide a forecast for the campaign, and we leverage the likely starting location and examine the range in CESM-LE conditions to contextualize the MOSAiC campaign. As the Polarstern searches for an initial floe from which to establish camp, the CESM-LE ensemble mean indicates that there is likely to be widespread ice cover with a mix of predominantly new, thin ice and some old, thick ice, but there is wide variability in the spatial ice coverage. Starting from the assumed likely starting location, the CESM-LE indicates that in Seasonal conditions a Transpolar Drift path is likely (47%), but would not have been as likely in Perennial conditions (27%). The increase in likelihood of a Transpolar Drift path is consistent with satellite derived tracks, which show the frequency of this type of trajectory increasing from 14% in the first half of the satellite record to 79% in the second half. The CESM-LE tracks show that in Seasonal conditions, as compared to Perennial conditions, thinner ice will drift more quickly (Morison & Goldberg, 2012; Rampal et al., 2009; Tschudi et al., 2019). The modelled Beaufort Gyre is stronger than observations due to biases in the atmospheric circulation (DeRepentigny et al., 2016), so the modelled tracks that enter the Beaufort Gyre may be due to a combination of thin ice and particularly strong atmospheric circulation in those ensemble members. There is a small (17%) chance the floe may melt out in August or September before a full calendar year, which was not the case for any observed or Perennial floes. Future campaigns could use climate model ensembles to better understand the likely conditions contributing to outlier, hazardous paths. These simulated paths can also be used to coordinate airborne or surface measurements with acquisition of satellite imagery. We find that in a Seasonal Arctic, the campaign may be visible by satellites by July, which is earlier than estimated using satellite observations or Perennial conditions. Ultimately, the path that MOSAiC takes can be used to validate and improve modelled sea ice motion (e.g. Beaufort Gyre strength and location).

MOSAiC observations will focus on coupled processes, so using a coupled climate model to analyse the representation of these coupled processes provides data that traditional weather or process modelling does not. While higher resolution coupled regional models can also provide information on modelled coupled processes, they do not have a direct linkage to extra-polar regions present in a global climate model important for understanding the exchanges between (both into and out of) the Arctic and elsewhere. To this end, we explore how the CESM-LE results can inform data collection during MOSAiC as well as how modelers can optimally use the data once the campaign is over to improve models. For the Seasonal tracks, there is a large range in possible initial sea ice conditions, so the campaign may start at the ice edge or well within the pack. The variability in melt season conditions is higher for the Seasonal than Perennial tracks, so identifying the initial sea ice state from which the campaign will start well in advance is less certain in Seasonal conditions due to the higher variability. A high-precision record

in one location for one sea ice type is just one realization in the wide range of possible sea ice states, so observations spanning a diverse range of sea ice conditions are crucial for improving modelled processes of the heterogeneous sea ice system. The initial MOSAiC camp will be established on a thick floe, which is likely to persist throughout the following year, though large variability in melt season sea ice conditions should be expected. We recommend both *in-situ* sampling of a large range in conditions as well as targeting airborne and spaceborne platforms on the anticipated drift tracks in the year before, during, and after the MOSAiC campaign to better understand the variability within the system. Because sea ice models represent subgrid-scale ice thickness categories and combines these into a coarser grid cell representation, it is particularly valuable to obtain observations spanning a range of ice conditions in order to enable process-oriented understanding and model representation over a fully representative distribution of sea ice conditions.

Regarding how models can best use the observational data once it is collected, we find that when comparing modelled data to a moving, observational point-based dataset that it is optimal is to use a Lagrangian framework. This framework allows us to analyse the processes virtual floes experience as they evolve in a similar way the observations on MOSAiC will measure processes impacting the real floe's evolution in time. Compared to the Lagrangian seasonal evolution, a regional average (e.g. polar-cap 70-90°N or Pan-Arctic) has a large discrepancy in the seasonal cycle. This is mainly due to differences in solar radiation for high-latitude floes near the pole.

We also used the model to identify interesting model behaviour that warrants further investigation. Using autocorrelations for the 30 unique initial floe conditions allowed us to explore how long there is predictability in the modelled sea ice system based on the initial sea ice state. While a thick initial sea ice floe for the campaign is likely to persist well into the following year, the emergence of predictability for ice thickness or concentration during the autumn freeze-up is not well understood and was unexpected. Observations of processes occurring during the autumn freeze-up taken during MOSAiC will be beneficial for understanding the formation and evolution of the sea ice thickness distribution and how this, in turn, affects sea ice predictability throughout the year. Future modelling work could explore other predictors (e.g. SST, remote influences) on the predictability of a sea ice floe, use metrics other than autocorrelation to quantify predictability, and include perfect model experiments (e.g. Blanchard-Wrigglesworth et al., 2011; Holland et al., 2013) initialized during the freeze up that might elucidate the mechanisms for this increase in autocorrelation.

**Code Availability**

The scripts used for this analysis in this paper can be found here: https://github.com/duvivier/MOSAIC_TC_2019.

**Data Availability**

The CESM-LE data are freely available: http://www.cesm.ucar.edu/projects/community-projects/LENS/.

**Author Contribution**

A.K. DuVivier designed the experiments, and A.K. DuVivier and P. DeRepentigny worked together to complete the Lagrangian Tracking. A.K. DuVivier make figures and performed the analysis on the experiments with input from P. DeRepentigny, M.M. Holland J.E. Kay. M. Webster and D. Perovich provided particular insight about how observational data and the model analysis could be optimally used together.

**Competing Interests**

The authors declare that they have no conflict of interest.

**Acknowledgements**

We greatly appreciate the input of MOSAiC organizers, in particular Thomas Krumpen and Matthew Shupe, in sharing their work to determine initial conditions and providing insight about the climate model tracks. This material is based upon work supported by the National Center for Atmospheric Research, which is a major facility sponsored by the National Science Foundation (NSF) under Cooperative Agreement No. 1852977. The CESM project is supported primarily by NSF, and computing and data storage resources, including the Cheyenne supercomputer (doi:10.5065/D6RX99HX), were provided by the Computational and Information Systems Laboratory (CISL) at NCAR. P. DeRepentigny acknowledges the support of the Natural Sciences and Engineering Council of Canada (NSERC), the Fond de recherche du Québec – Nature et Technologies (FRQNT) and the Canadian Meteorological and Oceanographic Society (CMOS) through PhD scholarships. M. Holland acknowledges support from NSF (OPP 1724748); D. Perovich acknowledges support by NSF; J. Kay acknowledges support from NSF (CAREER 1554659). M. Webster is supported by the National Aeronautics and Space Administration's New (Early Career) Investigator Program in Earth Science.

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

|  | Satellite (all: 1988-2015) | Satellite (1988-2001) | Satellite (2002-2015) | Perennial CESM-LE | Seasonal CESM-LE |
|---|---|---|---|---|---|
| **Total # of Tracks** | 28 tracks | 14 tracks | 14 tracks | 30 tracks | 30 tracks |
| **Average Distance** | 2018 km | 1725 km | 2311 km | 2572 km | 2944 km |
| **Standard Deviation** | 416 km | 301 km | 290 km | 173 km | 196 km |
| **Maximum Distance** | 2719 km | 2225 km | 2719 km | 2915 km | 3379 km |
| **Minimum Distance** | 1339 km | 1339 km | 1719 km | 2104 km | 2594 km |
| **Melts before Calendar Year (any endpoint)** | 0.0 % | 0.0 % | 0.0 % | 0.0 % | 16.7 % |
| **North Pole Endpoint** | 28.6 % | 42.9 % | 14.3 % | 63.3 % | 33.3 % |
| **Transpolar Drift Endpoint** | 46.4 % | 14.3 % | 78.6 % | 26.7 % | 46.7 % |
| **Russian Endpoint** | 25.0 % | 42.9 % | 7.1 % | 3.3 % | 3.3 % |
| **Canadian Endpoint** | 0.0 % | 0.0 % | 0.0 % | 6.7 % | 16.7 % |

**Table 1:** Statistics for LITS modelled tracks using satellite observations and Seasonal and Perennial CESM-LE conditions. The different endpoint regions (illustrated on Figure 1) are defined with final latitude/longitude values as follows: the North Pole sector (Lat. ≥ 87°); the Transpolar Drift sector (Lat. < 87°, 35° > Lon. > -60°); the Russian sector (Lat. < 87°, 180° ≥ Lon. ≥ 35°); the Canadian sector (Lat. < 87°, 60° ≥ Lon. ≥ -180°).


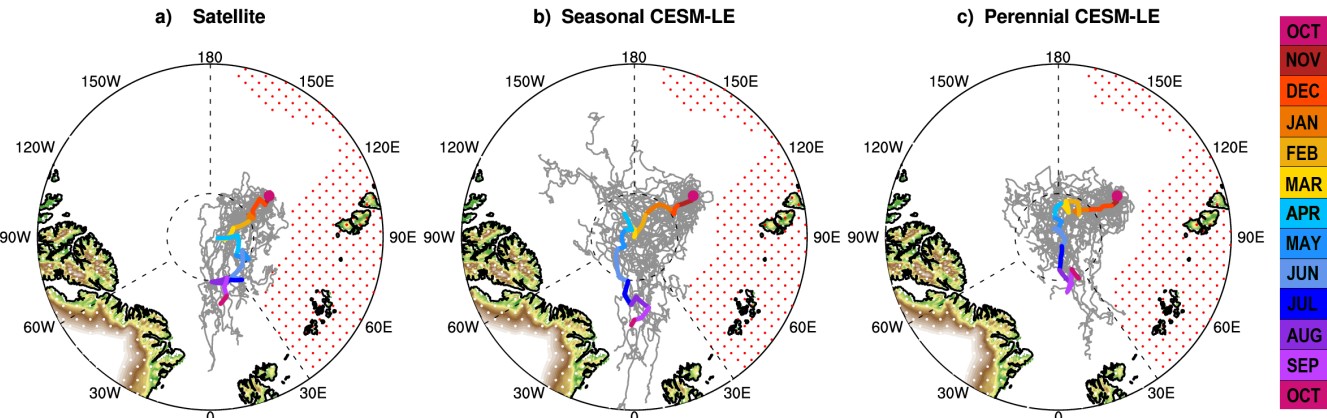

**Figure 1:** Unique ice floe tracks (grey) for satellite observations (a), Seasonal (b) and Perennial (c) CESM-LE ensembles. Representative tracks (coloured with corresponding months shown at far right) based on the individual tracks are overlain. Dashed lines indicate boundaries for the Transpolar Drift, North Pole, Russian, and Canadian sectors described in Table 1. The Russian exclusive economic zone (EEZ) is shown by red stipples.

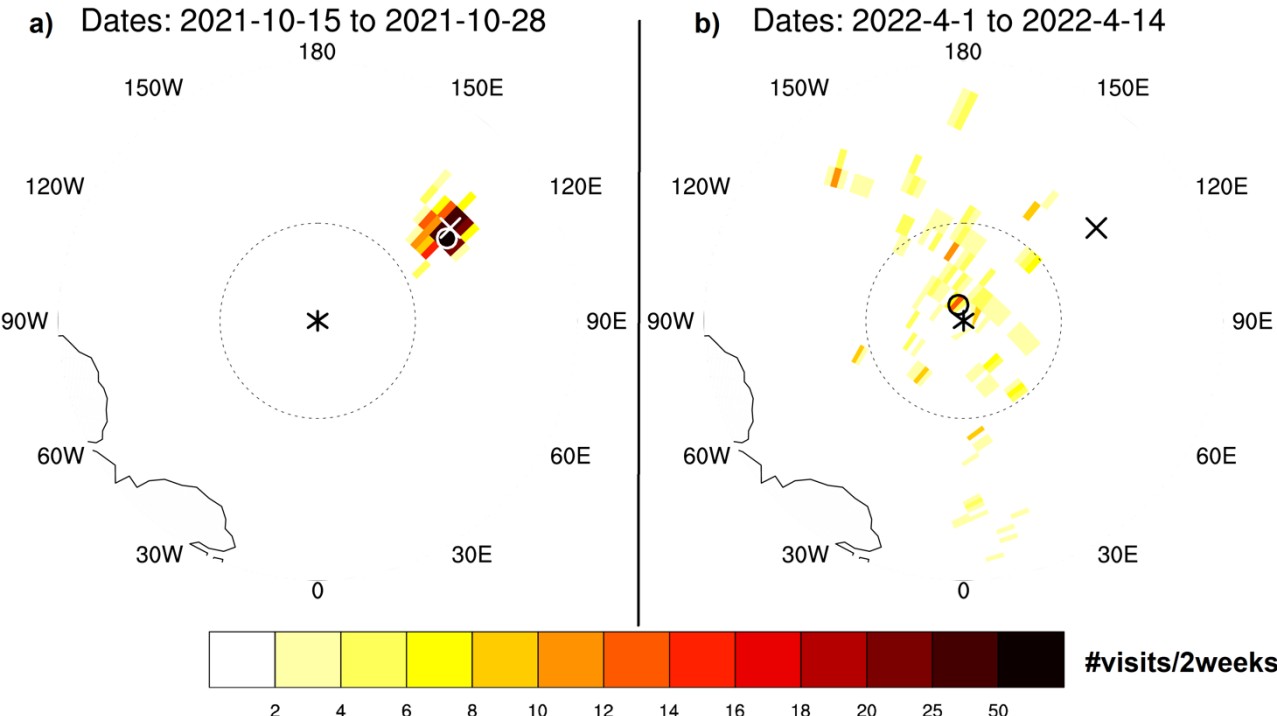

**Figure 2:** Counts of visits to each grid cell by Seasonal tracks in a two-week period during October 15 - October 28 (a) and April 1 - April 14 (b). The 'X' marks the starting location of the campaign, the circle marks the location of highest counts in the two-week time period, the star marks the North Pole. The dashed black line at 87°N shows, on average, where several current polar orbiting satellites used for cryospheric research begin to lose coverage due to satellite pole holes. A satellite's pole hole is unique to its orbit inclination and instrument swath.


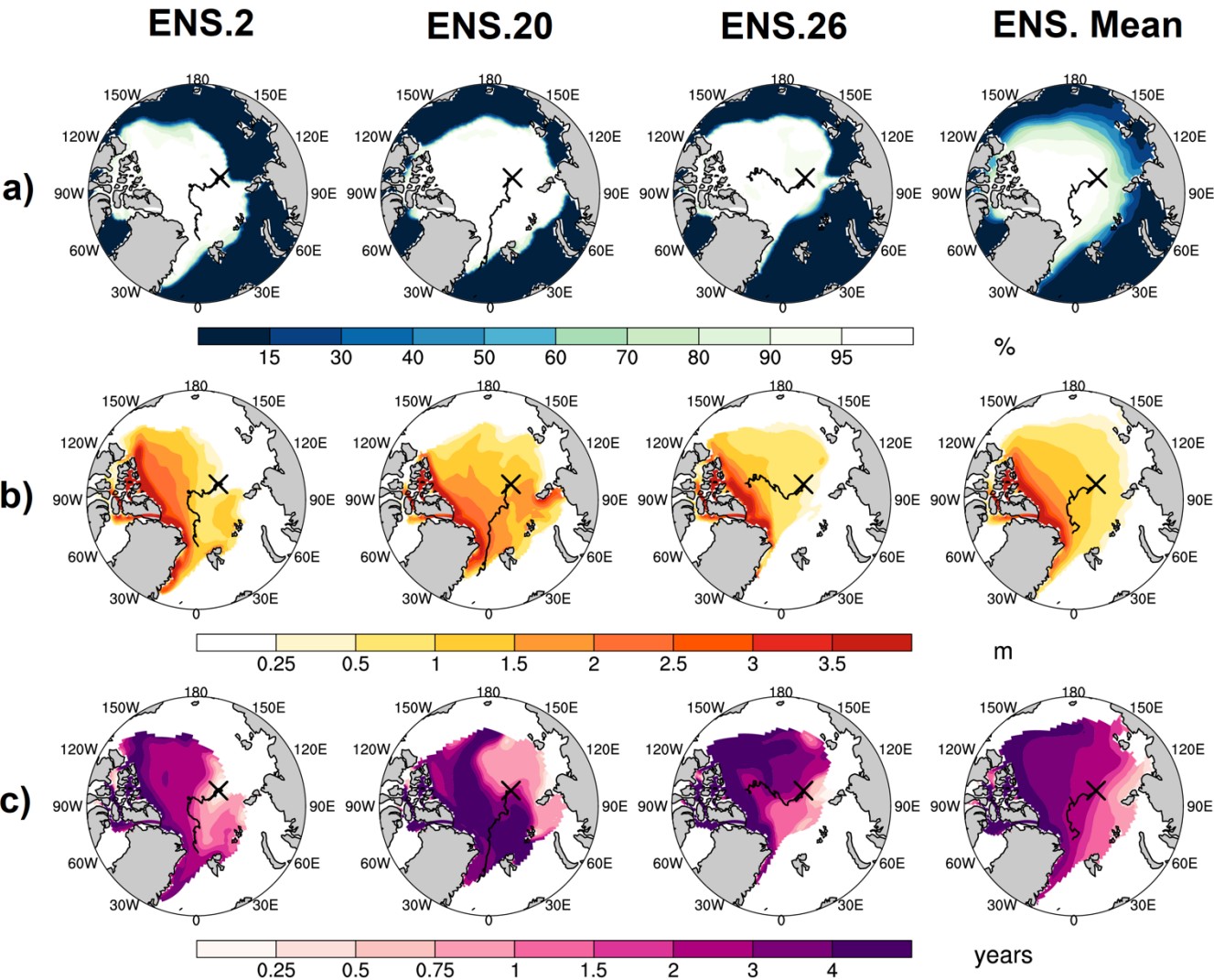

**Figure 3:** Seasonal grid cell daily mean sea ice concentration (a), thickness (b), and ice age (c) on October 15. Three random ensembles from the CESM-LE and the ensemble mean are shown. The black 'X' in each map denotes the campaign starting location. Overlain as black lines are the Lagrangian tracks for each ensemble member and the representative track from Fig.1 for the ensemble mean.

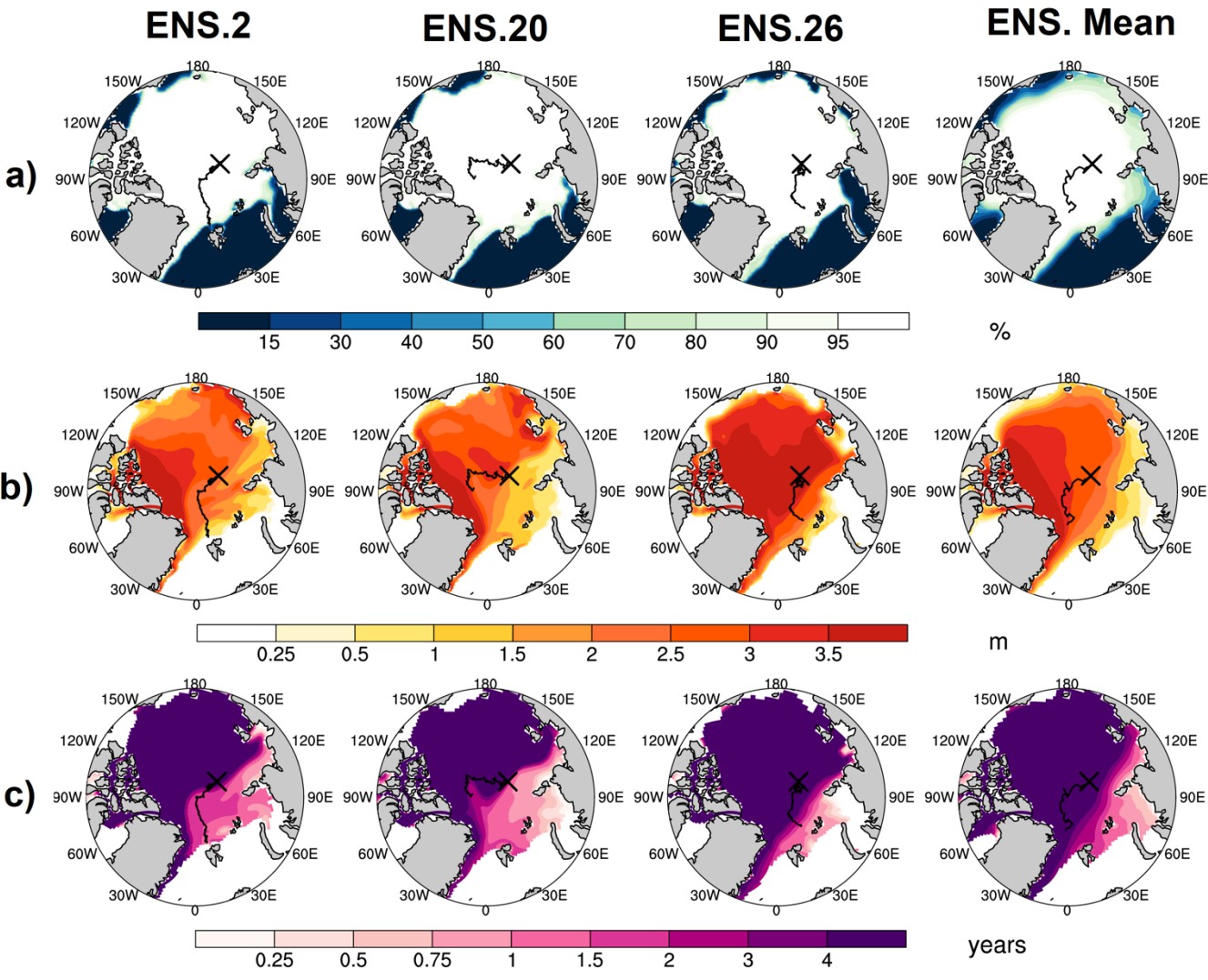

**Figure 4:** Perennial grid cell daily mean sea ice concentration (a), thickness (b), and ice age (c) on October 15. Three random ensembles from the CESM-LE and the ensemble mean are shown. The black 'X' in each map denotes the campaign starting location. Overlain as black lines are the Lagrangian tracks for each ensemble member and the representative track from Fig.1 for the ensemble mean.


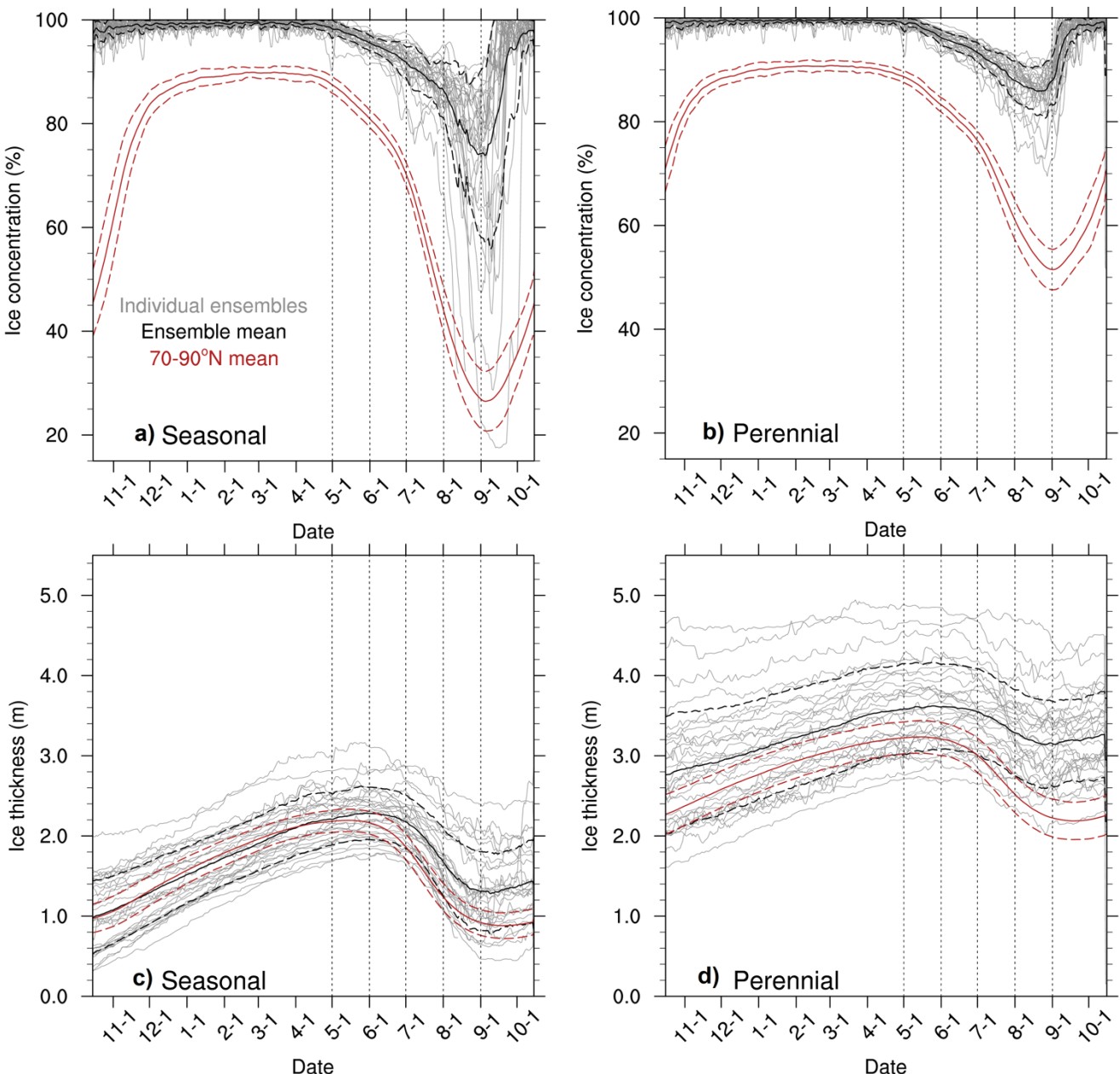

**Figure 5:** Year-long daily mean sea ice concentration (a,b) and sea ice thickness (c,d) for Seasonal (left column) and Perennial (right column) tracks. Ice state for each unique ice floe tracks (grey) are overlain by the ensemble mean (black, solid) and polar-cap (70-90°N) daily mean ice state (red, solid) and respective +/-1 standard deviation (dashed).

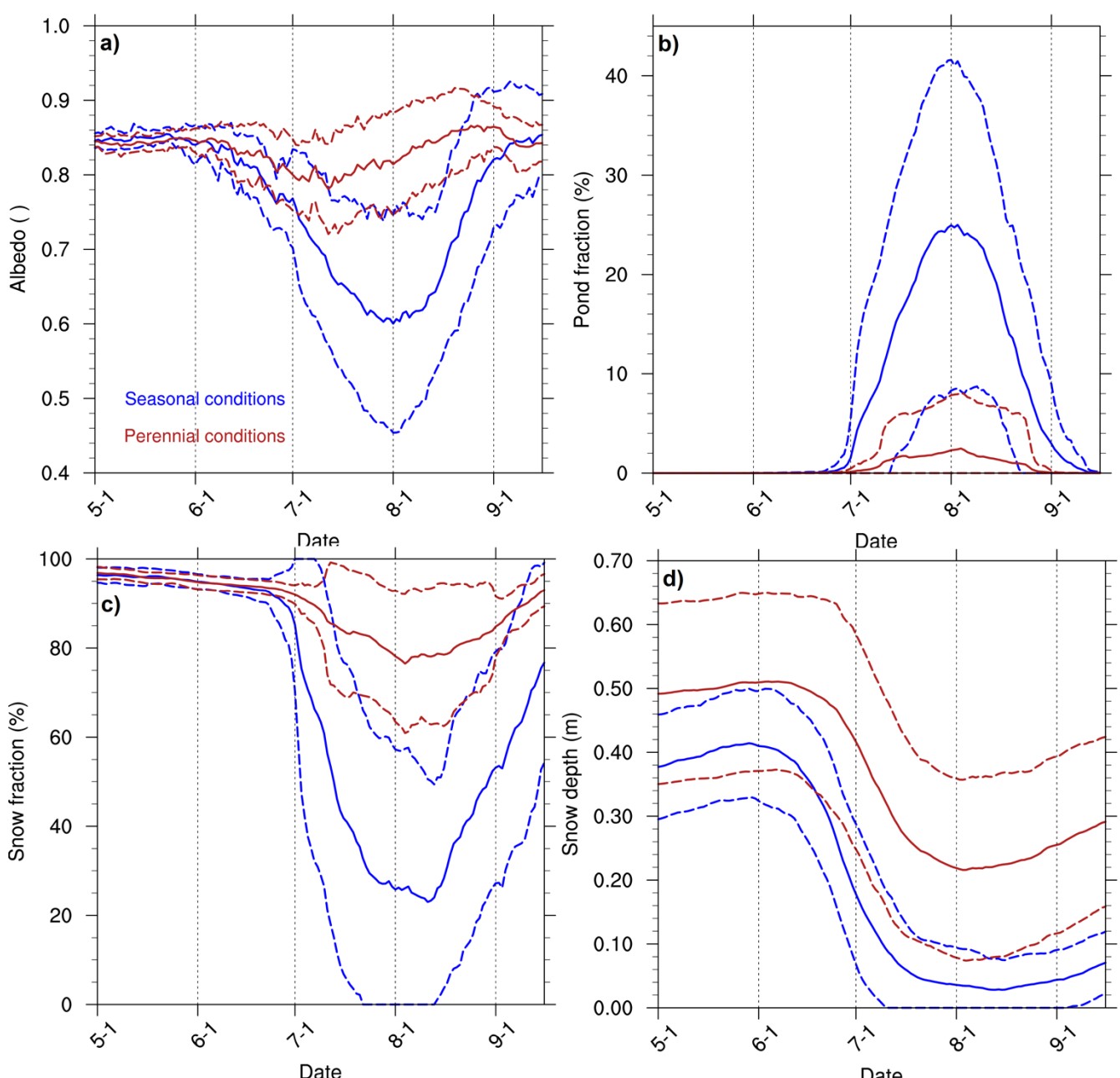

**Figure 6:** Melt season daily along-track mean sea ice albedo (a), pond fraction of model grid cell (b), snow fraction on sea ice (c), and snow depth on sea ice (d). The ensemble mean (solid) and +/-1 standard deviation envelope (dashed) are shown for Seasonal (blue) and Perennial (red) conditions. The melt season shown here corresponds to May 1 to September 15.


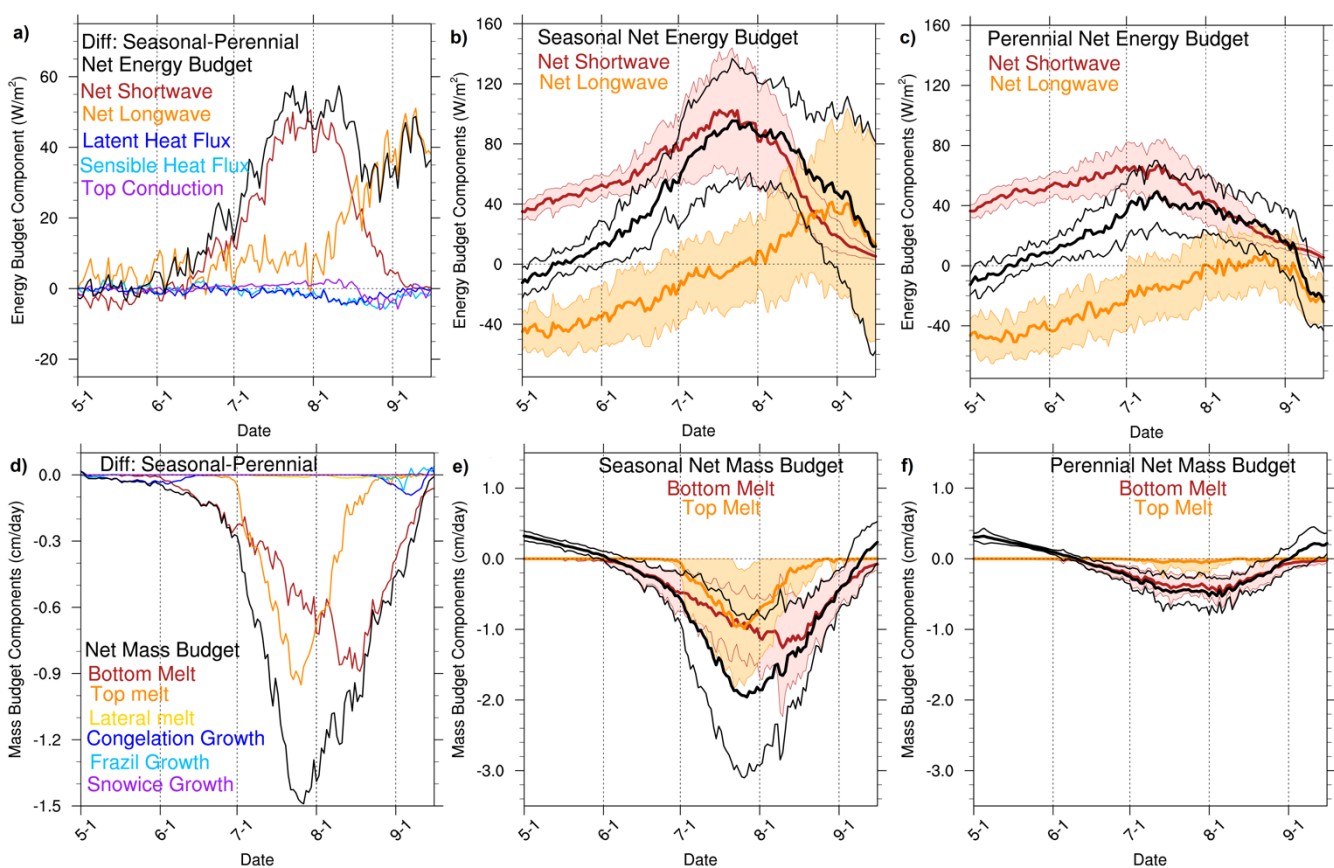

**Figure 7:** Melt season daily along-track mean surface energy budget (a) and mass budget (d) differences (Seasonal tracks minus Perennial tracks). The ensemble mean (solid) and +/-1 standard deviation envelope (shaded) are shown for the components with the largest differences for the energy budget (b, c) and mass budget (e, f). Components of the energy budget include: net (black), net shortwave (red), net longwave (orange), latent heat flux (dark blue), sensible heat flux (light blue), and top conductive flux (purple). Components of the mass budget include: net (black), bottom melt (red), top melt (orange), lateral melt (gold), congelation growth (dark blue), frazil growth (light blue), and snow-ice growth (purple). The melt season shown here corresponds to May 1 to September 15.

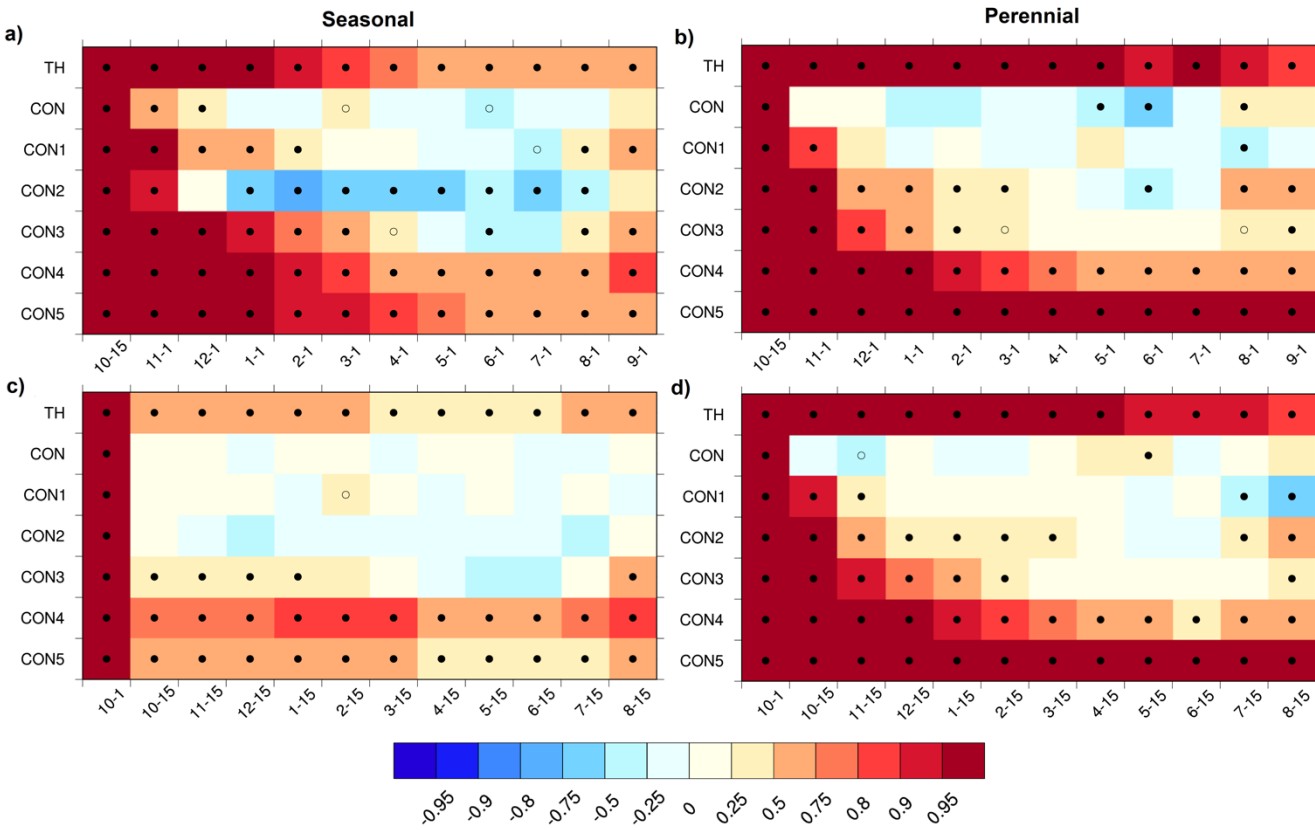


**Figure 8:** Correlation coefficients for sea ice variables between the initial condition and condition throughout the year for Seasonal (a, c) and Perennial (b, d) tracks initialized on October 15 (a, b; top row) and October 1 (c, d; bottom row). Solid (open) black dots indicate statistical significance at the 95% (90%) confidence level. From top to bottom, the variables shown are grid cell along-track mean ice thickness (TH), grid cell mean ice concentration (CON), and ice concentration for categories 1-5 (CON1-CON5).