# Peer review of "Going with the floe: tracking CESM Large Ensemble sea ice in the Arctic provides context for ship-based observations"

_The Cryosphere, 2019_

## Referee Comment (RC1) · Anonymous Referee #1 · 17 Jul 2019

This paper uses the CESM Large Ensemble to simulate possible floe tracks and floe thermodynamics throughout the course of MOSAiC. It suggests that the model tracks can assist with campaign planning and put the observations into context.

The paper is very well written, easy to follow, and provides clear graphics. Nevertheless, I unfortunately have severe doubts whether it really is suitable for publication in The Cryosphere. This is primarily because the relevance of the paper is somewhat unclear given its current framing.

1. If this really is meant as a guidance document to help campaign planning for MO-SAiC, it probably is best communicated to the MOSAiC planning staff rather than pub-

lished as a scientific paper. However, for this particular purpose, seasonal prediction systems initialized with the currently observed sea-ice state of the Arctic seem much better suited than a free simulation from a coupled climate model. It is my understanding that such system is in place to help the planning of MOSAiC, which by now is based on 45,000 individual simulations (see https://www.polarprediction.net/yopp-activities/sidfex/ for details).

2. The same holds for the assessment of "likely sea-ice conditions that the campaign will encounter during the year-long experiment". An initialized seasonal prediction system continuously updated until the start of MOSAiC likely provides more robust answers than a coupled, free simulation.

3. I must admit that I also failed to fully grasp the relevance of the discussed virtual floe track and floe evolution in a perennial ice cover. Maybe somewhat more discussion could be provided to explain why this discussion is included, given that the perennial ice cover discussed here no longer exists.

These are just my initial reactions after reading the paper. However, maybe (hopefully!) there's been a misunderstanding. In this case, I hope that the authors can sharpen the arguments for the overall framing of this paper.

---

## Referee Comment (RC2) · Anonymous Referee #2 · 22 Aug 2019

— Summary

DuVivier et al. present results that shed light on possible trajectories and corresponding ice conditions that the upcoming year-long Arctic drift campaign MOSAiC will encounter. Ensembles of trajectories are based mainly on CESM model data from different periods corresponding to past (perennial ice) versus present/near-future (seasonal ice) climate states, and to some extent also on satellite-derived drift data from the past few decades. The authors compute probabilities for various scenarios, such as to end up in the Beaufort Gyre, and present some conclusions that are potentially relevant for the planning of the campaign.

[Figure]

The paper is well-written, scientifically solid, and contains appropriate high-quality figures. I have one main remark relating to the fact that the authors have failed to include reference to similar work carried out by MOSAiC planners, which should certainly be taken into account. Apart from that I have only relatively minor suggestions for how the paper can be improved in my view. Overall, I recommend publication in The Cryosphere subject to minor-to-major revisions.

— Main point

The authors should include reference to and relate their work to similar work that has been carried out by the MOSAiC planners. While this work has to my knowledge not been published in a peer-reviewed journal, some of it has been published as "grey literature" in the MOSAiC Implementation Plan, version April 2018 (see link below). In Section 4.2 "Drift Trajectory and Re-supply", results for trajectories based on satellite-derived drift, similar to the satellite approach presented by DuVivier et al., are presented and probabilities of drifting into the Russian EEZ, into the "pole hole", the probability of "melting out", the expected drift distance etc. for different start positions are derived and discussed.

The present study by DuVivier et al. adds additional insight, in particular by extending the methodology to an ensemble of model trajectories and by studying present versus past conditions, so the paper is certainly a valuable addition. Nevertheless I consider it very important - and fair toward the MOSAiC planners - to refer to their work and to relate the present results to those contained in the MOSAiC Implementation Plan.

MOSAiC Implementation Plan: https://www.mosaic-expedition.org/fileadmin/user_upload/MOSAiC/Documents/MOSAiC_Implementation_Plan_April2018.pdf

— Other specific points

* There are some minor language errors distributed over the text, so I recommend a careful language check before final publication.
* P1L22: "We find that sea ice predictability emerges rapidly during the autumn freeze-up" (and corresponding parts of the main text related to predictability): By using only the autocorrelation relative to the initial state to quantify this, only the type of predictability related to the persistence of each single quantity is accounted for. However, predictability that is carried into the future through persistence of other quantities, conditions at other locations, and/or any non-persistence type of predictability is neglected. I therefore suspect that the sudden jump in "predictability" from initial states at the beginning versus the middle of October might be an artifact caused by the use of autocorrelation to measure predictability. In other words, I could well imagine that such a jump would be absent or less pronounced if predictability were quantified by means of "perfect-model" ensemble simulations initialised at those times of the year. I am not suggesting to do such additional simulations, but I recommend to mention the limitations of autocorrelation to quantify predictability and to formulate the result of a "rapid emergence of predictability" more cautiously.

* P1L30: Can't the reference to personal communication be replaced by a usual reference? There should be numerous up-to-date sources of the pan-Arctic sea ice extent that can be cited.

* P2L47: "These observations ...": which?

* P3L88-89: "For the satellite-derived drifts, more recent years tend to have longer drift distances (not shown)" (and other statements related to the first versus second half of the satellite period): I suggest to add two more columns to the table to show results for the first and second half of the satellite period.

* Second half of rows of Table 1: It would be more straight forward to compare the numbers if they were given as "percent" instead of number of tracks.

* The authors might consider adding the supplementary figure to the main paper; I think the figure is sufficiently interesting for the main part, and then no supplementary material would be required anymore.

\* P4L119-120: "when the field experiment might be observable by satellites": The size of the pole hole is different for different satellites, which might be worthwile mentioning.

\* The way they are computed by connecting modal values of spatial probability distributions, how different are the "most likely tracks" from tracks one would get if the ensemble centroids were used? Also, I am wondering how robust those modal-value based tracks are? E.g., in the example shown in Fig. S1-b, there are three distinct maxima, so the location of the modal value might easily jump from one such "hot spot" to another, no?

\* Related to the former point, one could relatively easily extend the sample size of the model trajectories by using a few years before and after the single years currently used for "Seasonal" and "Perennial"; the "climate change" occuring over a few years is arguably not too significant. Is there a particular reason why the authors have not extended the sample size in such a way?

\* P6-7L182-184: "By calculating the autocorrelation coefficient between the 30 unique initial floe conditions and the subsequent conditions each following month throughout the year, we are able to explore how long there is predictability in the sea ice system based on the initial sea ice state": Here you mention that this measures only predictability "based on the initial sea ice state", but I recommend to state more explicitly that, by capturing the initial state much more completely and considering not only correlation, predictability metrics based on initialised ensembles (e.g. perfect-model type) could well be considerably higher and show a weaker or no jump in predictability from early to mid October initial conditions.

\* P7L184-185: "In the CESM, the sea ice model represents subgrid-scale heterogeneity for five thickness categories": I suggest to include this and some other relevant information about the model, in particular the sea-ice component, in the "Data and Methods" section.

\* P7L187-194: Could you comment on the significant negative correlations obtained in
particular for CONC2, Seasonal, Oct.15?

* I assume that the probabilities highlighted in the "Discussion and Conclusions" section are quite sensitive to slight model biases, in particular in terms of the Arctic ice drift pattern. I would thus recommend to include a corresponding note of caution and to also provide the corresponding numbers derived from the satellite-based drift (possibly only the second half of the period), which might serve to give a rough idea about uncertainty in these numbers.

* P8L234-235: "the emergence of predictability during the autumn freeze-up is not well understood and was unexpected": see my corresponding remarks above.

* Figures 1,2,3,6,S1: A general recommendation is to use colour-blind friendly colour bars. ColorBrewer is an excellent source for such colour bars.

[Figure]

---

## Author Comment (AC1) · 18 Sep 2019

Thanks to the reviewers for their helpful comments that have improved this manuscript. Please see attached document for our response to Referee #1. In this document, responses to reviewers are in italics. Changes to text or figure captions are denoted by bold italics and the line numbers given refer to the modified manuscript that does not include tracked changes.

Please also note the supplement to this comment:
https://www.the-cryosphere-discuss.net/tc-2019-145/tc-2019-145-AC1-supplement.zip

---

## Author Comment (AC3) · 18 Sep 2019

Attached is the updated manuscript considering comments from both referees. I have included two versions: one is the "clean" manuscript and the second is the "tracked changes" manuscript.

Please also note the supplement to this comment:
https://www.the-cryosphere-discuss.net/tc-2019-145/tc-2019-145-AC3-supplement.zip

---

## Author Comment (AC5) · 25 Oct 2019

The attachment includes a revised version of the manuscript. One document has "track changes" and one is a "clean" version. This responses to the reviewers have been uploaded to each of their individual initial comments.

Please also note the supplement to this comment:
https://www.the-cryosphere-discuss.net/tc-2019-145/tc-2019-145-AC5-supplement.zip

---

## Author Response (AR1)

*Thanks to the reviewers for their helpful comments that have improved this manuscript. Responses to reviewers are in italics. Changes to text or figure captions are denoted by **bold italics** and the line numbers given refer to the modified manuscript that does not include tracked changes.*

\*\*\*\*\*\*\*\*\*\*\*\*\*\*\*\*\*\*\*\*\*\*\*\*\*\*\*\*\*\*\*\*\*\*\*\*\*\*\*\*\*\*\*\*\*\*\*\*\*\*\*\*\*\*\*\*\*\*\*

Anonymous Referee #1

This paper uses the CESM Large Ensemble to simulate possible floe tracks and floe thermodynamics throughout the course of MOSAiC. It suggests that the model tracks can assist with campaign planning and put the observations into context. The paper is very well written, easy to follow, and provides clear graphics. Nevertheless, I unfortunately have severe doubts whether it really is suitable for publication in The Cryosphere. This is primarily because the relevance of the paper is somewhat unclear given its current framing.

1. If this really is meant as a guidance document to help campaign planning for MOSAiC, it probably is best communicated to the MOSAiC planning staff rather than published as a scientific paper. However, for this particular purpose, seasonal prediction systems initialized with the currently observed sea-ice state of the Arctic seem much better suited than a free simulation from a coupled climate model. It is my understanding that such system is in place to help the planning of MOSAiC, which by now is based on 45,000 individual simulations (see https://www.polarprediction.net/yoppactivities/sidfex/ for details).

*We appreciate a great deal of work has gone into planning the MOSAiC campaign from determining a starting location to forecasting conditions, and we now better acknowledge that work within the manuscript. In particular, Thomas Krumpen of AWI has done a substantial amount of work to determine a likely starting location, and his suggested location was used as a foundational assumption for this paper. This is now more clearly acknowledged in the manuscript. We also do not intend this work to be a forecast of the actual conditions MOSAiC will encounter during its year-long drift, and the reviewer is correct that initialized forecasts would be the correct tool for that purpose. Our goals for this work are distinct from this guidance. The intent of this manuscript is instead a scientific analysis that provides context on possible measurements in light of internal variability and a changing climate state. We also demonstrate how free running climate model simulations can both be evaluated in a consistent way with observations as well as investigate ways in which a climate model might assist with future campaign planning. We have substantially revised the manuscript (as shown below in the bold text) to better reflect these goals throughout the paper. Because of the scientific focus, we believe that the manuscript is suitable for publication in The Cryosphere.*

*Our definition of planning is broad and taken to be providing information to the MOSAiC observational teams as they prepare to deploy and plan field strategies. In this sense, we have made deliberate efforts to communicate the results to the MOSAiC teams and work with them on particular questions they may have. This manuscript is primarily related to sea ice observations. Co-author Perovich is one of the Sea Ice Team leads for MOSAiC and co-author Webster is involved in Sea Ice remote sensing as part of MOSAiC and have been integral in some of the analysis presented because it is useful for their planning. In addition, these results have been presented at the NOAA ESRL laboratory in Boulder, CO to a number of MOSAiC participants*

and planners including Matt Shupe, who is one of the Atmospheric Team leads as well as a Coordination co-lead. Since the manuscript reviews have come back, we have been back in touch with several of the MOSAiC planning team members to be sure we can adequately reflect and reference their contributions, and we have made changes to the text to reflect these conversations.

However, the audience for this work is also the community at large, not just MOSAiC planners. We have added text contrasting an initial condition climate model ensemble with an initialized forecast to help clarify why we use the CESM-LE as a tool. The world we have observed is just one of many possible "worlds" in terms of the climate response to external forcing, and this is due to the inherent internal variability in the climate system. A free running climate model cannot and should not be expected to replicate observations since it is just one realization of possible climate response to external climate forcing. In past studies, observations were often compared directly with a single climate model experiment. Thus, we focus on the range of possible conditions and how the observations might fit into these as a way of more intelligently assessing and improving a free running climate model. In the past we have also seen numerous comparisons between SHEBA observations and polar-cap model averages, and here we investigate if this is an appropriate comparison (it is not). It is important to share these results with the wider community, some of whom will no doubt use MOSAiC observations when analyzing climate models.

As shown below, we have substantially changed the wording in the introduction and methods section and added to the discussion in order to 1) make the purpose of the study clear and distinct from other MOSAiC planning and 2) to clarify why the CESM-LE is the best tool for the analysis presented. We also now refer to the colored tracks shown on Fig. 1 as "representative" conditions rather than "likely" conditions to help clarify this is not a forecast track.

**Introduction – Lines 48-72**

[revised manuscript text omitted]

2. The same holds for the assessment of "likely sea-ice conditions that the campaign will encounter during the year-long experiment". An initialized seasonal prediction system continuously updated until the start of MOSAiC likely provides more robust answers than a coupled, free simulation.

*We believe this is a miscommunication about the purpose of this work. We are not providing a forecast, which we agree would be better informed by a tool like an initialized seasonal prediction system. However, we believe the free running climate model ensemble is another important tool that can provide different types of information relevant to the experiment or future, similar experiments. Please see the sections of the text highlighted in the response to point 1.*

3. I must admit that I also failed to fully grasp the relevance of the discussed virtual floe track and floe evolution in a perennial ice cover. Maybe somewhat more discussion could be provided to

explain why this discussion is included, given that the perennial ice cover discussed here no longer exists. These are just my initial reactions after reading the paper. However, maybe (hopefully!) there's been a misunderstanding. In this case, I hope that the authors can sharpen the arguments for the overall framing of this paper.

*One purpose of this work is to use a climate model to provide context to the observations obtained during the campaign in Seasonal conditions. To fulfill this goal, it is important to use Perennial conditions to contrast Seasonal conditions. For example, if we examined Seasonal tracks only we would see that a number of paths enter the Beaufort Gyre but we would not know if that path became more common with time as the climate system responding to greenhouse gas forcing or if it was just a model bias. By contrasting with the Perennial conditions, we found that these tracks appear to be related to the climate system response, though we did find that even in Perennial conditions the Beaufort Gyre is too strong. Using climate models enables this type of analysis and is informative because it is possible to examine changes in the mean state as well as the distribution of conditions. This can be seen in the Figure below, where we could have distributions of conditions in the Perennial and Seasonal conditions and see if the distribution has shifted, broadened, etc. Changes to the distribution of conditions could impact both modelers' and observationalists' expectations for the sea ice and also may suggest ways in which observations would be most useful for improving models. To make the purpose of using the Perennial conditions clear we've included the following in the text.*

**Lines 109-111**: **The purpose of including Perennial conditions, which no longer exist in the present-day Arctic, is to act as a contrast to the Seasonal conditions in terms of how the mean state and variability, or spread in conditions, has changed over time.**

*Figure (Courtesy of Alexandra Jahn, University of Colorado Boulder)*

[Figure]
* * *
Anonymous Referee #2

— Summary

DuVivier et al. present results that shed light on possible trajectories and corresponding ice conditions that the upcoming year-long Arctic drift campaign MOSAiC will encounter. Ensembles of trajectories are based mainly on CESM model data from different periods corresponding to past (perennial ice) versus present/near-future (seasonal ice) climate states, and to some extent also on satellite-derived drift data from the past few decades. The authors compute probabilities for various scenarios, such as to end up in the Beaufort Gyre, and present some conclusions that are potentially relevant for the planning of the campaign.

The paper is well-written, scientifically solid, and contains appropriate high-quality figures. I have one main remark relating to the fact that the authors have failed to include reference to similar work carried out by MOSAiC planners, which should certainly be taken into account. Apart from that I have only relatively minor suggestions for how the paper can be improved in my view. Overall, I recommend publication in The Cryosphere subject to minor-to-major revisions.

— Main point

The authors should include reference to and relate their work to similar work that has been carried out by the MOSAiC planners. While this work has to my knowledge not been published in a peer-reviewed journal, some of it has been published as "grey literature" in the MOSAiC Implementation Plan, version April 2018 (see link below). In Section 4.2 "Drift Trajectory and Re-supply", results for trajectories based on satellite derived drift, similar to the satellite approach presented by DuVivier et al., are presented and probabilities of drifting into the Russian EEZ, into the "pole hole", the probability of "melting out", the expected drift distance etc. for different start positions are derived and discussed.

The present study by DuVivier et al. adds additional insight, in particular by extending the methodology to an ensemble of model trajectories and by studying present versus past conditions, so the paper is certainly a valuable addition. Nevertheless I consider it very important - and fair toward the MOSAiC planners - to refer to their work and to relate the present results to those contained in the MOSAiC Implementation Plan. MOSAiC Implementation Plan: https://www.mosaicexpedition.org/fileadmin/user_upload/MOSAiC/Documents/MOSAiC_Implementation_Plan_April2018.pdf

*The reviewer is absolutely correct that substantial work was done by the MOSAiC planning team to identify an ideal starting location. In particular, Thomas Krumpen's suggested starting location was used as a foundational assumption for this work. Since the manuscript reviews have come back, we have been back in touch with several of the MOSAiC planning team members to be sure we can adequately reflect and reference their contributions. As a result, we have made changes to the text to reflect these conversations, the implementation document, and other data recently made available online about preliminary analyses they performed. From these communications we have determined that some of the analysis presented here evolved parallel but separately to similar work the MOSAiC planning team performed (e.g. co-author DeRepentigny has done previous work on ice floe movement between EEZs and co-author Webster is interested in satellite pole hole timing). We have modified the text extensively to better reflect the work the MOSAiC planners have done and acknowledge that their conclusion of a likely starting point was foundational for this work. Additionally, we have clarified that the*

*purpose of this study is not to provide a forecast, which may have been poorly communicated before, as another way to distinguish what we present here from the extensive analysis the MOSAiC planners have performed.*

**Introduction – Lines 48-72**

**In autumn 2019, the icebreaker RV Polarstern will be frozen into the Siberian Arctic with the aim of traversing the Transpolar Drift current over the following year. Extensive analysis using historical satellite data with the IceTrack Lagrangian approach (Krumpen et al., 2019) has been performed in order to identify MOSAiC's starting location, assist with logistical planning, and coordinate research efforts (Krumpen, 2019). Throughout the duration of the experiment dynamical sea ice forecasts, initialized with and assimilating the most up to date ice and weather data, will be performed for the particular MOSAiC track (Kauker, 2019). Analyses of an observationally-initialized ensemble forecast can provide skilful forecasts for MOSAiC conditions and information about how long these forecasts are skilful before the system diverges from the initial state. These types of in-depth observational and observationally-initialized forecast analyses are necessary and important for a successful campaign (IASC, 2016).**

**In recent years there has been increasing awareness of the impact of internal climate variability on the possible range of sea ice conditions and the resulting representativeness of a single year of observations ( Swart et al., 2015; Jahn et al., 2016). In this study, we use data from the Community Earth System Model (CESM) Large Ensemble (CESM-LE) project (Kay et al., 2015). The CESM-LE is an initial condition ensemble, meaning that each ensemble member represents one possible response of the climate system to the external forcing given inherent internal climate variability. The ensemble mean of the individual model experiments represents the response to the changing external forcing, whereas the difference of each ensemble member from to the ensemble mean provides a measure of internal variability. Using the Lagrangian Ice Tracking System (LITS; DeRepentigny et al., 2016), we derive the tracks of virtual sea ice floes for each ensemble member and the evolution of floe conditions over a calendar year. This analysis is not equivalent to examining observational tracks over time. Observational tracks are affected by both internal variability and forced change. Instead, the CESM-LE is an ideal tool for disentangling the effects of internal climate variability from forced change on the conditions a field campaign might encounter.**

**The purpose of this study is not to provide a forecast for the particular sea ice conditions during MOSAiC. Instead, we use the likely starting condition determined by the MOSAiC planners to address three fundamental goals: (1) offer insight on the representativeness of MOSAiC observations given the range of internal climate variability; (2) provide guidance about what types of observations can best assist with model improvement and appropriate ways these observations can be used to improve climate models; (3) show how free running climate model simulations might assist with future campaign planning.**

**Line 99: …the point 85°N, 125°E on October 15 (provided by Thomas Krumpen as a likely starting point for the MOSAiC campaign) for…**

**Lines 305-306 (in Acknowledgements): We greatly appreciate the input of MOSAiC organizers, in particular Thomas Krumpen and Matthew Shupe, in sharing their work to determine initial conditions and providing insight about the climate model tracks.**

— Other specific points

\* There are some minor language errors distributed over the text, so I recommend a careful language check before final publication.

*We have gone through the manuscript to check for language errors and have fixed many throughout the text.*

\* P1L22: "We find that sea ice predictability emerges rapidly during the autumn freezeup" (and corresponding parts of the main text related to predictability): By using only the autocorrelation relative to the initial state to quantify this, only the type of predictability related to the persistence of each single quantity is accounted for. However, predictability that is carried into the future through persistence of other quantities, conditions at other locations, and/or any non-persistence type of predictability is neglected. I therefore suspect that the sudden jump in "predictability" from initial states at the beginning versus the middle of October might be an artifact caused by the use of autocorrelation to measure predictability. In other words, I could well imagine that such a jump would be absent or less pronounced if predictability were quantified by means of "perfect-model" ensemble simulations initialised at those times of the year. I am not suggesting to do such additional simulations, but I recommend to mention the limitations of autocorrelation to quantify predictability and to formulate the result of a "rapid emergence of predictability" more cautiously.

*This is a good point and we have better clarified the limits of autocorrelation as a measure of predictability. We believe that it is relevant and useful to understand the persistence of ice conditions from the possible start location of a field campaign but appreciate that this is not necessarily the true "predictability" of conditions (and now mention this within the text). Note that we did examine the impact of other co-located quantities (i.e. SST on thickness, thickness on concentration, etc.) and did not find any clear relationships for the individual floes. We still find it interesting and surprising that there is such a large difference in the autocorrelation of thickness or concentration with its initial state for just a two-week period depending only if the floes start on Oct.1 vs. Oct.15 (Fig8a and 8c, top two rows). We have modified the discussion to highlight ways these questions could be explored in the future.*

*__Lines 219-223:__ By calculating the autocorrelation coefficient between the 30 unique initial floe conditions and the subsequent conditions each following month throughout the year, we are able to explore how long the initial sea ice state persists. This provides information on predictability of conditions, although notably there are other factors, such as the persistence of conditions that affect the sea ice state, that can give rise to predictability and are not accounted for in the autocorrelation analysis shown here.*

*__Lines 293-302:__ We also used the model to identify interesting model behaviour that warrants further investigation. Using autocorrelations for the 30 unique initial floe conditions allowed us to explore how long there is predictability in the modelled sea ice system based on the initial sea ice state. While a thick initial sea ice floe for the campaign is likely to persist well into the following year, the emergence of predictability for ice thickness or concentration during the autumn freeze-up is not well understood and was unexpected. Observations of processes occurring during the autumn freeze-up taken during MOSAiC will be beneficial for understanding the formation and evolution of the sea ice thickness distribution and how this, in turn, affects sea ice predictability throughout the year. Future modelling work could explore other predictors (e.g. SST, remote influences) on the predictability of a sea ice floe, use metrics other than autocorrelation to quantify predictability, and include perfect model experiments (e.g. Blanchard-Wrigglesworth et al., 2011; Holland et al., 2013) initialized during the freeze up that might elucidate the mechanisms for this increase in autocorrelation.*

* P1L30: Can't the reference to personal communication be replaced by a usual reference? There should be numerous up-to-date sources of the pan-Arctic sea ice extent that can be cited.

*We have updated this reference since the publication has now been released*

**Line 30: …recorded in the past twelve years (Richter-Menge et al., 2019).**

**Richter-Menge, J. A., Osborne, E., Druckenmiller, M., & Jeffries, M. O. (Eds.). (2019). The Arctic [in "State of the Climate in 2018"]. Bulletin of the American Meteorological Society, 100(9), S141–S168.**

* P2L47: "These observations ...": which?

*We have clarified this sentence:*

**Lines 46-48: MOSAiC aims to assess coupled air-sea-sea ice processes as well as to investigate the impact on ecosystems and biogeochemistry of the changing system in order to answer MOSAiC's driving question and improve our understanding and modelling of polar processes in a changing climate (Dethloff et al., 2016).**

* P3L88-89: "For the satellite-derived drifts, more recent years tend to have longer drift distances (not shown)" (and other statements related to the first versus second half of the satellite period): I suggest to add two more columns to the table to show results for the first and second half of the satellite period.

* Second half of rows of Table 1: It would be more straight forward to compare the numbers if they were given as "percent" instead of number of tracks.

*We have change Table 1 to have additional rows for the two satellite periods as well as converted to using percentages.*

*Table 1:*

|  | Satellite (all: 1988-2015) | Satellite (1988-2001) | Satellite (2002-2015) | Perennial CESM-LE | Seasonal CESM-LE |
|:---:|:---:|:---:|:---:|:---:|:---:|
| **Total # of Tracks** | 28 tracks | 14 tracks | 14 tracks | 30 tracks | 30 tracks |
| **Average Distance** | 2018 km | 1725 km | 2311 km | 2572 km | 2944 km |
| **Standard Deviation** | 416 km | 301 km | 290 km | 173 km | 196 km |
| **Maximum Distance** | 2719 km | 2225 km | 2719 km | 2915 km | 3379 km |

| Minimum Distance | 1339 km | 1339 km | 1719 km | 2104 km | 2594 km |
|---|---|---|---|---|---|
| Melts before Calendar Year (any endpoint) | 0.0 % | 0.0 % | 0.0 % | 0.0 % | 16.7 % |
| North Pole Endpoint | 28.6 % | 42.9 % | 14.3 % | 63.3 % | 33.3 % |
| Transpolar Drift Endpoint | 46.4 % | 14.3 % | 78.6 % | 26.7 % | 46.7 % |
| Russian Endpoint | 25.0 % | 42.9 % | 7.1 % | 3.3 % | 3.3 % |
| Canadian Endpoint | 0.0 % | 0.0 % | 0.0 % | 6.7 % | 16.7 % |

\* The authors might consider adding the supplementary figure to the main paper; I think the figure is sufficiently interesting for the main part, and then no supplementary material would be required anymore.

*We have added the supplementary Figure as Fig.2 and changed the subsequent figure numbers accordingly.*

\* P4L119-120: "when the field experiment might be observable by satellites": The size of the pole hole is different for different satellites, which might be worthwile mentioning.

*We changed the language accordingly in the text and in the caption for the newly added figure 2 to explicitly state that satellite pole holes are dependent on their orbit inclination and instrument swath. The satellite examples given show a wide range in coverage (from ~82.5°N for CloudSat to ~89.3°N for AMSR-E). For demonstration purposes, we show the average coverage of these examples in Figure 2 and state this explicitly in the caption. Our choice of satellites was partly based on their wide range in coverage, but also because these specific satellite products are fundamental for the MOSAiC projects co-author Webster will lead as PI.*

*Lines 153-160: Tracks in close proximity to the North Pole are often not observable by most polar orbiting satellites due to orbit inclinations and instrument swath creating gaps in coverage. These "pole holes" range in size (e.g., ~82.5°N for CloudSat; ~88°N for ICESat-2; ~89.3°N for AMSR-E) and are dependent on a satellite's orbit. Knowing when the floe is likely to be in these areas is valuable for planning and coordinating surface-based and airborne measurements to fill the high-latitude satellite "gap" (Fig. 2). Equally, knowing where and when the track is likely to emerge from satellite pole holes in spring after polar day has returned is valuable for planning visible image acquisition, such as that from DigitalGlobe's WorldView satellites (https://www.satimagingcorp.com/satellite-sensors/), to support operational and scientific needs.*

***Figure 2 caption**: Counts of visits to each grid cell by Seasonal tracks in a two-week period during October 15 - October 28 (a) and April 1 - April 14 (b). The 'X' marks the starting location of the campaign, the circle marks the location of highest counts in the two-week time period, the star marks the North Pole. The dashed black line at 87°N shows, on average, where several current polar orbiting satellites used for cryospheric research begin to lose coverage due to satellite pole holes. A satellite's pole hole is unique to its orbit inclination and instrument swath.*

\* The way they are computed by connecting modal values of spatial probability distributions, how different are the "most likely tracks" from tracks one would get if the ensemble centroids were used? Also, I am wondering how robust those modal-value based tracks are? E.g., in the example shown in Fig. S1-b, there are three distinct maxima, so the location of the modal value might easily jump from one such "hot spot" to another, no?

*The reviewer points out an important point here about distinct maxima and the possibility of "jumps" between maxima. We did not permit a path that had unphysical jumps between maxima. We have modified the text to call the tracks "representative" paths as opposed to "likely" paths and added clarification about how they should be used:*

***Lines 146-162**: These maps are also used to identify "representative" tracks (Fig.1, coloured lines) by identifying locations with high track counts that also formed a continuous path – i.e. a path in which unphysical "jumps" in the track were not permitted. It is important to note this is not a forecast of the likely path MOSAiC will take, but instead is meant to represent a reasonable path given the individual tracks from ensemble members under the same climate forcing and how these may differ between Seasonal and Perennial conditions. While all representative tracks follow a Transpolar Drift trajectory, the representative Seasonal path is longer and shifted further towards the Canadian Arctic compared to the representative observed and simulated Perennial paths, which end further north (Fig. 1). … All three representative tracks enter the satellite "gap" in December, but the difference in when the representative tracks exit the "gap" differs between September for observational and Perennial tracks and July for Seasonal tracks.*

\* Related to the former point, one could relatively easily extend the sample size of the model trajectories by using a few years before and after the single years currently used for "Seasonal" and "Perennial"; the "climate change" occuring over a few years is arguably not too significant. Is there a particular reason why the authors have not extended the sample size in such a way?

*The reviewer makes a good point here about how to extend the ensemble, but we did not do this because of concerns about computing resources, both CPU time and data output sizes. In order to do this experiment, we required more frequent model output than was typical for the CESM-LE. Therefore, we selected periods that represented the "seasonal" and "perennial" conditions and for which we had model restart files available. For these periods we ran the model a total of 2 years with daily sea ice output. The study could be extended in the future by running more years (or other ways, like an initialized ensemble) if additional computer resources were available. We did try extending the ensemble with the available data by trying different starting locations and dates, but the results were not significantly different than those shown here and so we just show the single starting location and date for simplicity of conveying our results. We have clarified our choice of these years in the text:*

*Lines 111-113: The specific years of model data were chosen based on model restart file availability needed to obtain daily values of all sea ice variables necessary for analysis as well as a clear representation and contrast of Perennial and Seasonal conditions.*

* P6-7L182-184: "By calculating the autocorrelation coefficient between the 30 unique initial floe conditions and the subsequent conditions each following month throughout the year, we are able to explore how long there is predictability in the sea ice system based on the initial sea ice state": Here you mention that this measures only predictability "based on the initial sea ice state", but I recommend to state more explicitly that, by capturing the initial state much more completely and considering not only correlation, predictability metrics based on initialised ensembles (e.g. perfect-model type) could well be considerably higher and show a weaker or no jump in predictability from early to mid October initial conditions.

*We have modified the discussion to highlight ways these questions could be explored in the future.*

*Lines 293-302: We also used the model to identify interesting model behaviour that warrants further investigation. Using autocorrelations for the 30 unique initial floe conditions allowed us to explore how long there is predictability in the modelled sea ice system based on the initial sea ice state. While a thick initial sea ice floe for the campaign is likely to persist well into the following year, the emergence of predictability for ice thickness or concentration during the autumn freeze-up is not well understood and was unexpected. Observations of processes occurring during the autumn freeze-up taken during MOSAiC will be beneficial for understanding the formation and evolution of the sea ice thickness distribution and how this, in turn, affects sea ice predictability throughout the year. Future modelling work could explore other predictors (e.g. SST, remote influences) on the predictability of a sea ice floe, use metrics other than autocorrelation to quantify predictability, and include perfect model experiments (e.g. Blanchard-Wrigglesworth et al., 2011; Holland et al., 2013) initialized during the freeze up that might elucidate the mechanisms for this increase in autocorrelation.*

* P7L184-185: "In the CESM, the sea ice model represents subgrid-scale heterogeneity for five thickness categories": I suggest to include this and some other relevant information about the model, in particular the sea-ice component, in the "Data and Methods" section.

*We have moved the information about the ice thickness distribution to the Data and Methods section:*

*Lines 103-106: CICE4, the sea ice model used in the CESM-LE, uses an ice thickness distribution to represent subgrid-scale heterogeneity which allows us to consider the predictability of concentration by thickness category. In the CESM-LE, we use five thickness categories that correspond to the following thickness ranges: 0-0.59 m, 0.6-1.39 m, 1.4-2.39 m, 2.4-3.59 m, 3.6+ m (Hunke & Lipscomb, 2008).*

* P7L187-194: Could you comment on the significant negative correlations obtained in particular for CONC2, Seasonal, Oct.15?

*The negative correlation has to do with ice growth over the winter and how ice shifts between categories in the ice thickness distribution. What this means is that if there is a high fraction of ice in category 2 (0.6-1.39m) during the start of the freeze up that in subsequent months there is less ice likely to be in this category because it will have become thicker and therefore fall into category 3, 4, or 5. We have added a line to the text to clarify about this.*

*Lines 227-229: The negative correlations for ice in category 2 (0.6-1.39 m) are due to the nature of the ice thickness distribution and indicates that ice that is initially in this category is likely to move out of this category to thicker categories throughout the winter.*

\* I assume that the probabilities highlighted in the "Discussion and Conclusions" section are quite sensitive to slight model biases, in particular in terms of the Arctic ice drift pattern. I would thus recommend to include a corresponding note of caution and to also provide the corresponding numbers derived from the satellite-based drift (possibly only the second half of the period), which might serve to give a rough idea about uncertainty in these numbers.

*We have added to the discussion a note about the known CESM-LE circulation bias as well as numbers based on the satellite- based tracks.*

*Lines 253-265: As the Polarstern searches for an initial floe from which to establish camp, the CESM-LE ensemble mean indicates that there is likely to be widespread ice cover with a mix of predominantly new, thin ice and some old, thick ice, but there is wide variability in the spatial ice coverage. Starting from the assumed likely starting location, the CESM-LE indicates that in Seasonal conditions a Transpolar Drift path is likely (47%), but would not have been as likely in Perennial conditions (27%). The increase in likelihood of a Transpolar Drift path is consistent with satellite derived tracks, which show the frequency of this type of trajectory increasing from 14% in the first half of the satellite record to 79% in the second half. The CESM-LE tracks show that in Seasonal conditions, as compared to Perennial conditions, thinner ice will drift more quickly (Morison & Goldberg, 2012; Rampal et al., 2009; Tschudi et al., 2019). The modelled Beaufort Gyre is stronger than observations due to biases in the atmospheric circulation (DeRepentigny et al., 2016), so the modelled tracks that enter the Beaufort Gyre may be due to a combination of thin ice and particularly strong atmospheric circulation in those ensemble members. There is a small (17%) chance the floe may melt out in August or September before a full calendar year, which was not the case for any observed or Perennial floes. Future campaigns could use climate model ensembles to better understand the likely conditions contributing to outlier, hazardous paths.*

\* P8L234-235: "the emergence of predictability during the autumn freeze-up is not well understood and was unexpected": see my corresponding remarks above.

*See comments above for P6-7L182-184.*

\* Figures 1,2,3,6,S1: A general recommendation is to use colour-blind friendly colour bars. ColorBrewer is an excellent source for such colour bars.

*We have modified the figures the reviewer lists to make them more color blind friendly.*

Figure 1

[Figure]

**Figure 2**

[Figure]

**Figure 3**

[Figure]

**Figure 4**

[Figure]

**Figure 7**

[revised manuscript text omitted]

Left

| Page 16: [3] Formatted Table | Alice DuVivier | 10/29/19 7:18:00 AM |
|---|---|---|

Formatted Table

| Page 16: [4] Formatted | Alice DuVivier | 10/29/19 7:18:00 AM |
|---|---|---|

Centered

| Page 16: [5] Inserted Cells | Alice DuVivier | 10/29/19 7:18:00 AM |
|---|---|---|

Inserted Cells

| Page 16: [6] Formatted | Alice DuVivier | 10/29/19 7:18:00 AM |
|---|---|---|

Centered

| Page 16: [7] Inserted Cells | Alice DuVivier | 10/29/19 7:18:00 AM |
|---|---|---|

Inserted Cells

| Page 16: [8] Formatted | Alice DuVivier | 10/29/19 7:18:00 AM |
|---|---|---|

Centered

| Page 16: [9] Inserted Cells | Alice DuVivier | 10/29/19 7:18:00 AM |
|---|---|---|

Inserted Cells

| Page 16: [10] Formatted | Alice DuVivier | 10/29/19 7:18:00 AM |
|---|---|---|

Centered

| Page 16: [11] Formatted | Alice DuVivier | 10/29/19 7:18:00 AM |
|---|---|---|

Font: 12 pt

| Page 16: [12] Formatted | Alice DuVivier | 10/29/19 7:18:00 AM |
|---|---|---|

Centered

| Page 16: [13] Formatted Table | Alice DuVivier | 10/29/19 7:18:00 AM |
|---|---|---|

Formatted Table

| Page 16: [14] Formatted | Alice DuVivier | 10/29/19 7:18:00 AM |
|---|---|---|

Centered

| Page 16: [15] Inserted Cells | Alice DuVivier | 10/29/19 7:18:00 AM |
|---|---|---|

Inserted Cells

| Page 16: [16] Formatted | Alice DuVivier | 10/29/19 7:18:00 AM |
|---|---|---|

Centered

**Page 16: [18] Formatted**      **Alice DuVivier**      **10/29/19 7:18:00 AM**

Centered

**Page 16: [19] Formatted Table**      **Alice DuVivier**      **10/29/19 7:18:00 AM**

Formatted Table

**Page 16: [20] Formatted**      **Alice DuVivier**      **10/29/19 7:18:00 AM**

Centered

**Page 16: [21] Formatted**      **Alice DuVivier**      **10/29/19 7:18:00 AM**

Centered

**Page 16: [22] Formatted**      **Alice DuVivier**      **10/29/19 7:18:00 AM**

Centered

**Page 16: [23] Formatted**      **Alice DuVivier**      **10/29/19 7:18:00 AM**

Centered

**Page 16: [24] Formatted Table**      **Alice DuVivier**      **10/29/19 7:18:00 AM**

Formatted Table

**Page 16: [25] Inserted Cells**      **Alice DuVivier**      **10/29/19 7:18:00 AM**

Inserted Cells

**Page 16: [26] Formatted**      **Alice DuVivier**      **10/29/19 7:18:00 AM**

Centered

**Page 16: [27] Formatted**      **Alice DuVivier**      **10/29/19 7:18:00 AM**

Centered

**Page 16: [28] Formatted Table**      **Alice DuVivier**      **10/29/19 7:18:00 AM**

Formatted Table

**Page 16: [29] Formatted**      **Alice DuVivier**      **10/29/19 7:18:00 AM**

Centered

**Page 16: [30] Formatted**      **Alice DuVivier**      **10/29/19 7:18:00 AM**

Centered

**Page 16: [31] Formatted Table**      **Alice DuVivier**      **10/29/19 7:18:00 AM**

Formatted Table

**Page 16: [32] Deleted**      **Alice DuVivier**      **10/29/19 7:18:00 AM**

**Page 16: [32] Deleted**          **Alice DuVivier**          **10/29/19 7:18:00 AM**

▾

**Page 16: [32] Deleted**          **Alice DuVivier**          **10/29/19 7:18:00 AM**

▾

**Page 17: [33] Deleted**          **Alice DuVivier**          **10/29/19 7:18:00 AM**

▾

**Page 17: [33] Deleted**          **Alice DuVivier**          **10/29/19 7:18:00 AM**

▾

**Page 17: [33] Deleted**          **Alice DuVivier**          **10/29/19 7:18:00 AM**

▾

**Page 17: [33] Deleted**          **Alice DuVivier**          **10/29/19 7:18:00 AM**

▾

**Page 17: [33] Deleted**          **Alice DuVivier**          **10/29/19 7:18:00 AM**

▾

**Page 17: [33] Deleted**          **Alice DuVivier**          **10/29/19 7:18:00 AM**

▾

**Page 17: [33] Deleted**          **Alice DuVivier**          **10/29/19 7:18:00 AM**